# LINC00665: An Emerging Biomarker for Cancer Diagnostics and Therapeutics

**DOI:** 10.3390/cells11091540

**Published:** 2022-05-04

**Authors:** Chenming Zhong, Zijun Xie, Jinze Shen, Yunhua Jia, Shiwei Duan

**Affiliations:** 1Department of Clinical Medicine, School of Medicine, Zhejiang University City College, Hangzhou 310015, China; 196002141@nbu.edu.cn (C.Z.); shenjz@stu.zucc.edu.cn (J.S.); 2Medical Genetics Center, School of Medicine, Ningbo University, Ningbo 315211, China; 196002100@nbu.edu.cn

**Keywords:** LINC00665, cancer, diagnosis, prognosis, signaling pathway, ceRNA

## Abstract

Long intergenic noncoding RNA 00665 (LINC00665) is located on human chromosome 19q13.12. LINC00665 was upregulated in eighteen cancers and downregulated in two cancers. LINC00665 not only inhibits 25 miRNAs but also directly affects the stability of ten protein-coding genes. Notably, LINC00665 also encodes a micro-peptide CIP2A-BP that promotes triple-negative breast cancer progression. LINC00665 can participate in five signaling pathways to regulate cancer progression, including the Wnt/β-catenin signaling pathway, TGF-β signaling pathway, NF-κB signaling pathway, PI3K/AKT signaling pathway, and MAPK signaling pathway. Aberrant expression of LINC00665 in breast cancer, gastric cancer, and hepatocellular carcinoma can be used for disease diagnosis. In addition, aberrant expression of LINC00665 is closely associated with clinicopathological features and poor prognosis of various cancers. LINC00665 is closely associated with the effects of anticancer drugs, including gefitinib and cisplatin in non-small cell lung cancer, gemcitabine in cholangiocarcinoma, and cisplatin-paclitaxel in breast cancer. This work systematically summarizes the diagnostic and prognostic values of LINC00665 in various tumors, and comprehensively analyzes the molecular regulatory mechanism related to LINC00665, which is expected to provide clear guidance for future research.

## 1. Introduction

Long noncoding RNAs (lncRNAs) are generally defined as transcripts greater than 200 nucleotides in length with limited or no protein-coding capacity. LncRNAs are involved in various physiological and pathological processes at the cellular level through multiple regulatory mechanisms, including biological processes such as proliferation, differentiation, stemness, migration, invasion, and apoptosis in cancer development [1].

Long intergenic noncoding RNA 00665 (LINC00665) is a newly discovered lncRNA [2] located on human chromosome 19q13.12. Aberrant expression of LINC00665 is prevalent in human tumor diseases. Since its first report in 2018 [3], LINC00665 is found to be aberrantly expressed in more than 10 cancers. Among them, the expression of LINC00665 can not only be used to distinguish tumors from the adjacent tissues, but also is closely related to clinical characteristics and poor prognosis of patients.

Competing endogenous RNA (ceRNA) generally refers to the way lncRNAs regulate target mRNAs by sponging the corresponding miRNAs [4]. The ceRNA network centered on LINC00665 involves 25 miRNAs. Meanwhile, LINC00665 can also directly target the interaction of 10 protein-coding genes. LINC00665 encodes a micro-peptide CIP2A-BP with a molecular weight of 52 Daltons [5]. CIP2A-BP can directly bind the tumor oncogene CIP2A to replace the B56γ subunit of PP2A, thereby inhibiting the invasion and metastasis of triple-negative breast cancer [5]. Furthermore, LINC00665 is involved in the regulation of at least five signaling pathways.

Abnormally upregulated LINC00665 was also involved in drug resistance in non-small cell lung cancer (gefitinib and cisplatin) [6,7] and cholangiocarcinoma (gemcitabine) [8]. In breast cancer, LINC00665 also acts as an independent predictor of the efficacy of cisplatin-paclitaxel combination therapy [9].

Current evidence suggests that LINC00665 is a very promising biomarker and target for cancer research. Although there are many related reports on LINC00665, a systematic summary is lacking. Therefore, this work summarizes the diagnostic and prognostic values of aberrant expression of LINC00665 in cancer. In addition, this work comprehensively analyzes the molecular regulatory pathways related to LINC00665 and looks forward to the research directions and challenges of LINC00665, which is expected to provide hints for future related research.

## 2. Aberrant Expression of LINC00665 in Human Tumors

As an emerging oncogenic factor, abnormal expression of LINC00665 exists in 18 types of cancers, including tumors in multiple human reproductive, digestive, respiratory, neurological, and endocrine systems (Table 1). LINC00665 is highly expressed in 18 cancers, including breast cancer [9,10,11,12,13], prostate cancer [14,15,16], ovarian cancer [17,18,19], cervical cancer [20], endometrial cancer [21], gastric cancer [22,23,24,25], hepatocellular carcinoma [3,26,27], colorectal cancer [8,28,29,30], non-small cell lung cancer [6,7,31], lung adenocarcinoma [1,32,33,34,35], glioma [36], thymic epithelial tumor [37], osteosarcoma [2,38], multiple myeloma [39], melanoma [40], acute myeloid leukemia [41], and acute T lymphoblastic leukemia [42].

However, in triple-negative breast cancer [5], the expression of LINC00665 was significantly lower than in normal cells. In triple-negative breast cancer, activated TGF-β/Smad4 signaling significantly increases the mRNA level of 4E-BP1, which inhibits the translation of the micro-peptide CIP2A-BP by binding it to eIF4F [5]. CIP2A-BP is a micro-peptide directly encoded by LINC00665 [5]. In the future, we need to verify whether translational repression of CIP2A-BP is the key to the reduced expression of LINC00665 [5].

Furthermore, in gliomas, aberrant expression of LINC00665 has two opposing observations. Ruan et al. proposed that LINC00665 could act as a protective factor for gliomas and inhibit the malignant development of tumors through the TAF15|LINC00665/MTF1|YY2/GTSE1 axis [43]. Dai et al. found that LINC00665 was upregulated as an oncogenic factor and promoted glioma progression through a ceRNA mechanism (LINC00665/miR-34a-5p/AGTR1) [36]. These two studies used human normal astrocytes as cell line controls, which were purchased from ScienCell Research Laboratories [43] and YaJi Biological [36], respectively. In addition, Dai et al. found that LINC00665 [36] was abnormally upregulated by analyzing five glioma cell lines (U87 MG, LN229, A172, U373, and U251); while Ruan et al. used only two glioma cell lines (U251 and U87). These small differences may contribute to the divergence of LINC00665 expression in gliomas. Further determination of the molecular mechanism of LINC00665 in glioma is required in the future.

As shown in Table 1, qRT-PCR and RNA sequencing (RNA-seq) [9,13,16] are commonly used to determine the expression of LINC00665 [10]. The analysis methods of LINC00665 aberrant expression between groups include Student’s *t*-test [14,15], Analysis of variance (ANOVA) [12,13], and Chi-square test [9]. In addition, online bioinformatics analysis techniques, such as gene expression profiling interactive analysis (GEPIA) [33,41,42], were also used to analyze the aberrant expression of LINC00665 and the correlation between LINC00665 and various parameters, such as patient baseline information, clinicopathological features, prognosis, etc.

Functional experiments showed that high expression of LINC00665 could regulate cancer progression by affecting biological processes, such as cancer cell proliferation, apoptosis, migration, and invasion (Table 1). The normal physiological process of epithelial-mesenchymal transition (EMT) is crucial for embryonic development, tissue morphology, wound healing, and pathological conditions, such as fibrosis and tumor progression. The key to the development of EMT is the initiation of Vimentin and N-cadherin and the inhibition of E-cadherin, thereby enhancing cell migration and invasion ability. The high expression of LINC00665 is also closely related to EMT. In gastric cancer [24], breast cancer [11], ovarian cancer [44], and other cancers, abnormal upregulation of LINC00665 can promote the EMT process.

## 3. Association of LINC00665 with Clinicopathological Characteristics

High expression of LINC00665 was closely associated with clinicopathological features of 10 cancers (Table 2). In reproductive system tumors, LINC00665 was significantly positively associated with larger tumors and lymph node metastases in breast cancer [9,11,12,13], prostate cancer [15], and ovarian cancer [17]. In breast cancer [11,12,13] and prostate cancer [15], high expression of LINC00665 was also associated with the advanced TNM stage. In ovarian cancer, high expression of LINC00665 was associated with the Federation internationale of gynecologie and obstetrotrique (FIGO) stage. In high-grade serous ovarian cancer [19], high expression of LINC00665 was inversely correlated with macrophage and dendritic cell infiltration levels.

In digestive system tumors, highly expressed LINC00665 was associated with lymph node metastasis, low tumor differentiation, and advanced TNM staging in gastric cancer [23,24], hepatocellular carcinoma [3,26,27], colorectal cancer [28], and cholangiocarcinoma [8]. In gastric cancer, LINC00665 was positively correlated with the depth of tumor invasion [24]. In hepatocellular carcinoma, high expression of LINC00665 points to larger tumor size [27] and more severe vascular invasion [3].

In respiratory tumors, upregulated LINC00665 was significantly associated with larger tumors, advanced TNM stage, and lymph node metastasis in non-small cell lung cancer [7] and lung adenocarcinoma [1,32]. In nervous system tumors, LINC00665 was found to be positively correlated with the malignant pathological grade of glioma [43].

## 4. The Prognostic and Diagnostic Value of LINC00665

Aberrant expression of LINC00665 is associated with prognosis in cancer patients. As shown in Table 2, high expression of LINC00665 was significantly associated with decreased overall survival in 11 cancer patients, including breast cancer [11,13], prostate cancer [15,16], ovarian cancer [17,18,44], gastric cancer [24], hepatocellular carcinoma [3,26,27], cholangiocarcinoma [8], non-small cell lung cancer [7], lung adenocarcinoma [1,33,35], glioma [36], thymus epithelial tumor [37], and osteosarcoma [2,38]. Upregulated LINC00665 was also significantly associated with lower disease-free survival in patients with multiple cancers, including breast cancer [11], prostate cancer [16], ovarian cancer [17], cholangiocarcinoma [8], non-small cell lung cancer [7], and lung adenocarcinoma [1]. However, in triple-negative breast cancer patients, decreased expression of LINC00665 was associated with lower overall and disease-free survival [5].

In addition, the high expression of LINC00665 has a potential diagnostic value of cancer (Table 2). In breast cancer, ROC analysis demonstrated that LINC00665 could distinguish normal breast cells from diseased breast cells with an AUC value of 0.906 [9]. LINC00665 can also be used to differentiate gastric cancer tissues from normal adjacent tissues. In gastric cancer, ROC analysis showed that the AUC value of LINC00665 was 0.828, which indicated that LINC00665 could be used as a diagnostic marker for gastric cancer [24]. In hepatocellular carcinoma, LINC00665 can differentiate between normal and tumor tissues. ROC analysis showed that the AUC value of LINC00665 was 0.614, and the specificity and sensitivity were 0.53 and 0.55, respectively, indicating that LINC00665 has the potential to diagnose hepatocellular carcinoma [3].

The current study prefers to use LINC00665 as an independent diagnostic marker. LINC00665 is a promising diagnostic biomarker for breast cancer [9], gastric cancer [24], and hepatocellular carcinoma [3].

## 5. Signaling Pathways Associated with LINC00665 in Cancer

LINC00665 can regulate five signaling pathways, thereby promoting the occurrence and development of cancer. Signaling pathways associated with LINC00665 in cancer include the Wnt/β-catenin signaling pathway, TGF-β signaling pathway, NF-κB signaling pathway, PI3K/AKT signaling pathway, and MAPK signaling pathway (Figure 1).

### 5.1. Wnt/β-Catenin Signaling Pathway

Wnts are powerful regulators of cell proliferation and differentiation, and their signaling pathways involve proteins directly involved in gene transcription and cell adhesion. β-catenin is a central player in the Wnt pathway, functioning as a transcriptional cofactor and structural adaptor protein [45]. CTNNB1 is a key coactivator of the TCF/LEF family of transcription factors that mediate transcriptional activation of Wnt/β-catenin signaling in the nucleus [20].

In cervical cancer, after LINC00665 silencing, CTNNB1, a key factor in the Wnt pathway, was significantly downregulated, while the expression of DKK1, a Wnt pathway inhibitor, was significantly increased. This suggests that LINC00665 may promote the proliferation, migration, invasion, and EMT of HeLa cells, by activating the Wnt-CTNNB1/β-catenin signaling pathway [20].

In colorectal cancer, LINC00665 can interact with U2AF2, enhance the binding between U2AF2 and CTNNB1 mRNA, and increase the stability of CTNNB1 mRNA, thereby further activating the Wnt/β-catenin signaling pathway and stimulating the proliferation and invasion of colorectal cancer cells. Among them, U2AF2 is an RNA binding protein (RBP), which has a certain binding ability to LINC00665 and CTNNB1 mRNA [29].

A study in gastric cancer found that, with the silencing of LINC00665, Wnt signaling was inactivated in gastric cancer cells, and the expression of β-catenin and cyclinD1 was significantly inhibited [22]. In addition, knockdown of LINC00665 also downregulated Wnt/β-Catenin signaling and the expression of nuclear transcriptional regulator BCL9L in cholangiocarcinoma cells treated with gemcitabine [8].

### 5.2. TGF-β Signaling Pathway

Transforming growth factor-beta (TGF-β) can activate multiple signal transduction pathways by binding to cell surface receptors and is essential in many cellular processes, such as immunosuppression, growth inhibition, EMT, and cell invasion role [46]. In the late stages of cancer, TGF-β signaling increases the expression of mesenchymal markers N-cadherin and vimentin and downregulates the epithelial marker E-cadherin to promote EMT in tumor cells by mediating SMAD [47]. A study in gastric cancer found that silencing of LINC00665 can downregulate the expression levels of TGF-β and its downstream factors Smad-2 and α-SMA, indicating that LINC00665 may promote the progression of EMT in gastric cancer through the TGF-β/Smad-2 signaling pathway [24].

However, in triple-negative breast cancer, Smad4 inhibits micro-peptide synthesis of LINC00665 [5]. As a transcription factor for the translational repressor protein 4E-BP1, Smad4 significantly increases 4E-BP1 mRNA levels upon activation of the TGF-β/Smad signaling pathway [5]. The mRNA 4E-BP1 inhibits translation by binding to eukaryotic translation initiation factor-4F (eIF4F), thereby reducing the expression of the micro-peptide CIP2A-BP of LINC00665 [5].

### 5.3. NF-κB Signaling Pathway

Nuclear factor Kappa B (NF-κB) is widely recognized as a key regulator of inflammation, immunity, and cell survival, and plays an important role in human tumor progression [26]. Activation of NF-κB is a common event in hepatocellular carcinoma and is associated with a transformed phenotype at advanced stages of cancer. LINC00665, induced by NF-κB, enhances the activity of activated protein kinase (PKR) and maintains the stability of PKR by blocking ubiquitin/proteasome-dependent degradation of PKR. As an important regulator of NF-κB signal transduction, the stability of PKR is beneficial to NF-κB signal transduction, thus forming a positive feedback regulation system [26]. LINC00665 can significantly promote liver cancer cell proliferation and tumorigenicity in vitro and in vivo, and the discovery of the NF-κB/LINC00665/PKR/NF-κB positive feedback loop provides a new way to understand the link between inflammation and cancer [26]. In addition, NF-κB1-induced LINC00665 regulates neuronal inflammation and apoptosis induced by spinal cord injury by sponging miR-34a-5p, suggesting that the NF-κB1/LINC00665/miR-34a-5 axis may be an effective target for the treatment of spinal cord injury [48].

### 5.4. PI3K/AKT Signaling Pathway

The PI3K/Akt signaling pathway primarily mediates receptor-induced cell survival and is frequently altered in human cancers [49]. When PI3K binds to growth factor receptors, such as EGFR, it can catalyze the conversion of PIP2 into PIP3, which induces the translocation of Akt to the plasma membrane [50]. Akt is subsequently activated by PDK1 and mTORC2 complexes, which in turn target downstream by means of effectors to regulate metabolism, growth, and migration in mammalian cells [50]. In addition, the PI3K/Akt signaling pathway also regulates the development and stability of T cells [51].

In non-small cell lung cancer, LINC00665 upregulates EZH2, an important component of the initiation complex (PRC2), thereby activating the PI3K/AKT signaling pathway and mediating EGFR expression, weakening the inhibition of EGFR kinase by the drug gefitinib [6]. In T-cell acute lymphoblastic leukemia, LINC00665 can upregulate the protein levels of p-PI3K and p-AKT, suggesting that LINC00665 may enhance the viability, migration, and invasion of T-ALL cells by activating the PI3K/Akt signaling pathway [42].

In triple-negative breast cancer, activated AKT can play an important role in tumorigenesis and metastasis through downstream phosphorylation of NF-κB. As a tumor suppressor, the micro-peptide CIP2A-BP, encoded by LINC00665, can inhibit PI3K/AKT/NFκB signal transduction, thereby limiting the migration and invasion of triple-negative breast cancer cells [5].

### 5.5. MAPK Signaling Pathway

MAPKs are serine/threonine-protein kinases. Conventional MAPKs in mammals include c-Jun NH2-terminal kinase (JNK), P38 MAPK, and extracellular signal-regulated kinase (ERK) [52]. As an important part of the MAPK signaling pathway, the ERK signaling pathway is a major determinant of various cellular processes that control cell proliferation, survival, differentiation, and metastasis [53]. The ERK pathway promotes tumor cell proliferation and metastasis by stimulating the expression of matrix metalloproteinases and vimentin. Matrix metalloproteinases degrade the extracellular matrix to induce cell motility, and vimentin is an important protein in cell deformation and motility [53]. In lung adenocarcinoma, LINC00665, induced by transcription factor SP1, upregulates AKR1B10 by sponging miR-98-5p. In turn, overexpressed AKR1B10 promotes p-ERK1/2, MMP2, and vimentin expression, which mediate activation of the MAPK signaling pathway, thereby enhancing cancer cell proliferation and invasion in vitro and in vivo [1].

## 6. The ceRNA Network Centered on LINC00665

The ceRNA hypothesis describes a regulatory network among messenger RNAs, transcribed pseudogenes, and long noncoding RNAs that compete for miRNA binding [54]. CeRNA networks greatly expand functional genetic information in the human genome and play an important role in pathological conditions, such as cancer [55].

As shown in Figure 2, the ceRNA network centered on LINC00665 involves 25 miRNAs aberrantly expressed in 16 cancers, including miR-379-5p [10,25], miR-3619-5p [12,38], miR-551b-5p [13], miR-1224-5p [14], miR-34a-5p [17,36], miR-146a-5p [44], miR-449b-5p [19], miR-149-3p [23], miR-186-5p [27], miR-9-5p [28], miR-126-5p [30], miR-214-3p [29,39], miR-424-5p [8], miR-138- 5p [31], miR-98-5p [1], miR-181c-5p [33], miR-195-5p [34], miR-let-7b [35], miR-140 [37], miR-3199 [37], miR-708-5p [2], miR-142-5p [2], miR-224-5p [40], miR-4458-5p [41], and miR-101 [42].

In reproductive system tumors, LINC00665 accelerates breast cancer progression by sponging miR-379-5p [10], miR-3619-5p [12] and miR-551b-5p [13]. Among them, the LINC00665/miR-379-5p axis can upregulate LIN28B, an important regulator of EMT [10], while the LINC00665/miR-3619-5p axis can upregulate CTNNB1, a key protein coding-gene in the Wnt/β-catenin signaling pathway [12]. In ovarian cancer, the LINC00665/miR-34a-5p axis [17] and LINC00665/miR-146a-5p axis [44] promote cancer progression by upregulating E2F3 and CXCR4, respectively. LINC00665 can upregulate SND1 and RRAGD by sponging miR-1224-5p and miR-449b-5p, and promote the progression of prostate cancer [14] and high-grade serous ovarian cancer [19], respectively.

In digestive system tumors, the LINC00665/miR-186-5p/MAP4K3 axis and the LINC00665/miR-424-5p/BCL9L axis promote the progression of hepatocellular carcinoma [27] and cholangiocarcinoma [8], respectively. In gastric cancer, RNF2 and GRP78 are upregulated through the LINC00665/miR-149-3p axis [23] and LINC00665/miR-379-5p axis [25], respectively, to promote gastric cancer progression. In colorectal cancer, LINC00665 can sponge miR-9-5p [28], miR-126-5p [30] and miR-214-3p [29] to promote tumor growth, migration and invasion. Among them, ATF1 and CTNNB1 were upregulated by LINC00665/miR-9-5p axis and LINC00665/miR-214-3p axis, respectively. The downstream genes PAK2 and FZD3 can be upregulated through the LINC00665/miR-126-5p axis, and subsequently activate the Wnt/β-catenin signaling pathway, thereby promoting the proliferation of colorectal cancer cells.

In respiratory tumors, LINC00665 competitively binds to miR-138-5p, thereby upregulating E2F3 and promoting tumor growth and invasion [31]. In lung adenocarcinoma, LINC00665 promotes tumor growth by upregulating ZIC2, MYCBP and CCNA2 by sponging miR-181c-5p [33], miR-195-5p [34] and miR-let-5p [35], respectively. In addition, the LINC00665/miR-98-5p/AKR1B10 axis can activate the MAPK signaling pathway and its downstream gene MMPs, thereby promoting tumor cell invasion and cancer metastasis [1].

In glioma, LINC00665 can sponge miR-34a-3p, thereby upregulating AGTR1, promoting tumor growth and invasion [36]. In thymic epithelial tumors, LINC00665 promoted tumor growth and metastasis by competitively binding to miR-140 and miR-3199, upregulating MYO10 and WASF3, respectively [37]. In osteosarcoma, LINC00665 can upregulate RAP1B by sponging miR-708-5p and miR-142-5p and promote tumor cell proliferation, migration, and invasion [2]. In addition, LINC00665 competitively binds to miR-3619 and can also promote osteosarcoma progression [38]. In multiple myeloma, LINC00665 upregulates PSMD10 and ASF1B by sponging miR-214-3p, thereby promoting proliferation and inhibition of apoptosis of multiple myeloma cells [39]. In melanoma, the LINC00665/miR-224-5p axis elevates VMA21 expression and promotes tumor cell proliferation and migration [40]. In acute myeloid leukemia, upregulation of DOCK1, via the LINC00665/miR-4458-5p axis, activates Rac1 expression; thereby promoting cancer progression [41]. In addition, LINC00665 can compete with miR-101 to enhance the viability, migration, and invasion of cancer cells in T-cell acute lymphoblastic leukemia by activating the PI3K/Akt signaling pathway [42].

In conclusion, LINC00665 can regulate the expression of its downstream target genes through competitive binding with miRNAs, thereby acting as an oncogene in various cancers.

## 7. The Encoded Micro-Peptide of LINC00665 and the Protein-Coding Genes It Directly Targets

LINC00665 encodes a 52-dalton micro-peptide CIP2A-BP. In triple-negative breast cancer, CIP2A-BP acts as a tumor suppressor gene, inhibiting tumor invasion and metastasis by inactivating the PI3K/AKT pathway [5]. It is worth mentioning that this is the first report of CIP2A-BP. The discovery of CIP2A-BP provides a new direction for future research of LINC00665, which is expected to become a new target in cancer [5]. In addition, LINC00665 can directly target 10 protein-coding genes to regulate the occurrence and development of various cancers (Figure 3).

In glioma, the TAF15|LINC00665/MTF1|YY2/GTSE1 axis inhibits malignant tumor progression [43].TATA-box-binding protein-associated factor 15 (TAF15) is a member of the FET family and plays an important role in regulating mRNA transcription, RNA splicing, and trafficking. Both TAF15 and LINC00665 were downregulated in glioma carcinoma tissues and cancer cells. In cancer cells, overexpressed LINC00665 can reduce the stability of transcriptional regulators MTF1 and YY2, and downregulate the expression of microtubule localization protein 1 (GTSE1) in G2 and S phases, thereby attenuating the proliferation, migration, and invasion of glioma cells and apoptosis inhibition [43].

In prostate cancer, LINC00665 can interact with EZH2 and LSD1, recruiting them to the KLF2 promoter to repress its transcription, thereby promoting the malignant progression of cancer [15]. EZH2 and LSD1 are negative regulators of transcription through H3K27me3 and H3K4me2, respectively [15]. In addition, the LINC00665/EZH2/CDKN1 C axis enhances the proliferation, migration, and sensitivity of non-small cell lung cancer cells to the chemotherapeutic drug cisplatin (DDP) [7]. The high mobility group AT-hook 1 (HMGA1) is a non-histone chromatin-binding protein that is overexpressed in several tumor types and is associated with tumor invasion, metastasis, and drug resistance [56]. In endometrial cancer, LINC00665 can directly bind to the HMGA1 protein to promote tumor metastasis and invasion [21]. Y-box-binding protein 1 (YB-1) is a DNA/RNA-binding protein. In unstressed cells, YB-1 is mainly localized to the cytoplasm and regulates mRNA stabilization, splicing, and translation. Under stress conditions, the translocation of YB-1 to the nucleus activates the transcription of various genes involved in cancer progression and multidrug resistance [32]. In lung adenocarcinoma, LINC00665 can directly bind to YB-1 protein to enhance its stability, and the accumulated nuclear YB-1 activates the expression of ANGPT4, ANGPTL3, and VEGFA by binding to the promoter, facilitating tumor-associated angiogenesis in vitro and in vivo [32]. In addition, the potential target genes EDEM1, CAPNS1, SQSTM1, and SERPINA1 of LINC00665 may be involved in the autophagy process of ovarian cancer cells [18].

## 8. The Relationship between LINC00665 and the Efficacy of Anticancer Drugs

LINC00665 attenuates the efficacy of gefitinib [6] and cisplatin [7] in non-small cell lung cancer, and the efficacy of gemcitabine in cholangiocarcinoma [8]. Meanwhile, LINC00665 can be used as a predictor of the outcome of cisplatin-paclitaxel neoadjuvant therapy in breast cancer [9] (Figure 4).

Gefitinib, an epidermal growth factor receptor (EGFR) tyrosine kinase inhibitor [6], has been used in first-line treatment of EGFR-mutant non-small cell lung cancer and can significantly improve progression-free survival in patients [57]. LINC00665 is upregulated in non-small cell lung cancer and can activate the PI3K/AKT pathway by increasing the stability of EZH2, an important component of the initiation complex (PRC2), and mediates the expression of EGFR, thereby counteracting the effect of gefitinib [6].

Cisplatin is a first-line chemotherapeutic drug that kills cancer cells by inhibiting their DNA replication process and disrupting cell membrane structures, but long-term cisplatin treatment can lead to drug resistance and serious side effects [7]. In cisplatin treatment of non-small cell lung cancer, LINC00665 overexpression promotes cancer cell proliferation and induces chemoresistance to cisplatin [7].

Gemcitabine is a deoxycytidine analog that is incorporated into DNA duplexes during cellular DNA synthesis, preventing chain elongation and blocking cell cycle progression [58]. Intrinsic or acquired resistance to gemcitabine is frequently observed in patients with cholangiocarcinoma receiving gemcitabine as first-line chemotherapy [8]. Silencing LINC00665 reduced gemcitabine resistance and cell viability in gemcitabine-resistant cell lines HuCCT1-Gem and SNU-245-Gem, thereby enhancing the toxic effects of gemcitabine on cancer cells. Meanwhile, knockdown of LINC00665 inhibited the EMT and cell stemness of gemcitabine-resistant cells and downregulated Wnt/β-Catenin signaling and the expression of its nuclear transcriptional regulator BCL9L [8].

In addition, LINC00665 predicts response to cisplatin-paclitaxel neoadjuvant chemotherapy in patients with hormone receptor-positive breast cancer [9]. Cisplatin kills tumor cells by inducing DNA damage, while paclitaxel causes cell proliferation arrest and cell death, by inhibiting microtubule dynamics and activating mitotic checkpoints [59]. The results of both univariate and multivariate logistic regression analysis showed that the expression of LINC00665 was an independent predictor of pathological complete response (pCR) in breast cancer patients after cisplatin-paclitaxel neoadjuvant chemotherapy.

In conclusion, LINC00665 may be an important driver of cancer cell resistance to gefitinib, cisplatin, and gemcitabine, and has the potential to predict the efficacy of cisplatin-paclitaxel neoadjuvant chemotherapy.

## 9. Conclusions

LINC00665 is a tumor-promoting factor, newly discovered in recent years, which has great potential as a diagnostic and prognostic marker for various tumors and is expected to become a therapeutic target for various tumors. High expression of LINC00665 has potential diagnostic value in breast, gastric and hepatocellular carcinoma. High expression of LINC00665 is closely related to the clinicopathological features of 10 cancers, including tumor size, depth of invasion, lymph node metastasis, and TNM staging. In addition, abnormal expression of LINC00665 is also significantly correlated with the prognosis of cancer patients and the treatment outcomes of several chemotherapeutic drugs.

LINC00665 is involved in the regulation of at least five signaling pathways, including the Wnt/β-catenin signaling pathway, TGF-β signaling pathway, NF-κB signaling pathway, PI3K/AKT signaling pathway, and MAPK signaling pathway. By competitively binding 25 miRNAs, LINC00665 establishes a complex ceRNA network. The micro-peptide CIP2A-BP, encoded by LINC00665, can play an important inhibitory role in the progression of triple-negative breast cancer. Meanwhile, LINC00665 is involved in the regulatory process of six cancers by directly targeting ten protein-coding genes (Figure 5).

LINC00665 can enhance the resistance of tumor cells to various chemotherapeutic drugs, including the resistance of non-small cell lung cancer cells to gefitinib and cisplatin [6,7], and the resistance of cholangiocarcinoma cells to gemcitabine [8]. At the same time, LINC00665 can be used to predict the efficacy of cisplatin-paclitaxel in breast cancer patients [9].

Current studies suggest that LINC00665 is upregulated as an oncogene in most cancers, but some studies have found that in triple-negative breast cancer and glioma, LINC00665 can be downregulated as a tumor suppressor. LINC00665, which is generally upregulated in breast cancer, was found to be downregulated in triple-negative breast cancer. In breast cancer, upregulated LINC00665 promotes cancer progression by sponging 3 miRNAs (miR-379-5p, miR-3619-5p, miR-551b-5p). The LINC00665/miR-379-5p/LIN28B axis [10] and the LINC00665/miR-3619-5p/CTNNB1 axis [12] were involved in the regulation of EMT and Wnt/β-catenin signaling pathways, respectively. In addition, two studies [9,11] also demonstrated the upregulation of LINC00665 in breast cancer, but the mechanism of action of LINC00665 was not thoroughly explored. In triple-negative breast cancer, the protein CIP2A-BP encoded by LINC00665 can act as a protective factor to hinder cancer progression by inhibiting the PI3K/AKT/NFκB pathway [5]. In glioma, there are two opposing insights into the aberrant expression and mechanism of action of LINC00665. One view is the same as mainstream studies, that LINC00665 is upregulated as an oncogenic factor and promotes glioma progression through a ceRNA mechanism (LINC00665/miR-34a-5p/AGTR1). However, another view is that LINC00665 can act as a protective factor for glioma, inhibiting the malignant development of the tumor through the TAF15|LINC00665/MTF1|YY2/GTSE1 axis. The above inconsistency may be due to different research directions related to LINC00665 or the unclear regulatory mechanism of LINC00665 in glioma. In conclusion, our understanding of LINC00665 is limited, and more comprehensive studies are needed in the future.

In conclusion, LINC00665 is a promising lncRNA. In the future, it is necessary to further explore the regulatory mechanism of LINC00665 in different cancers and to establish a more refined LINC00665 regulatory network. At the same time, related research on the efficacy of LINC00665 on tumor therapeutic drugs could lay the foundation for clinically targeted therapy in cancer. In the future, it is necessary to broaden the role of LINC00665 in cancer.

## Figures and Tables

**Figure 1 cells-11-01540-f001:**
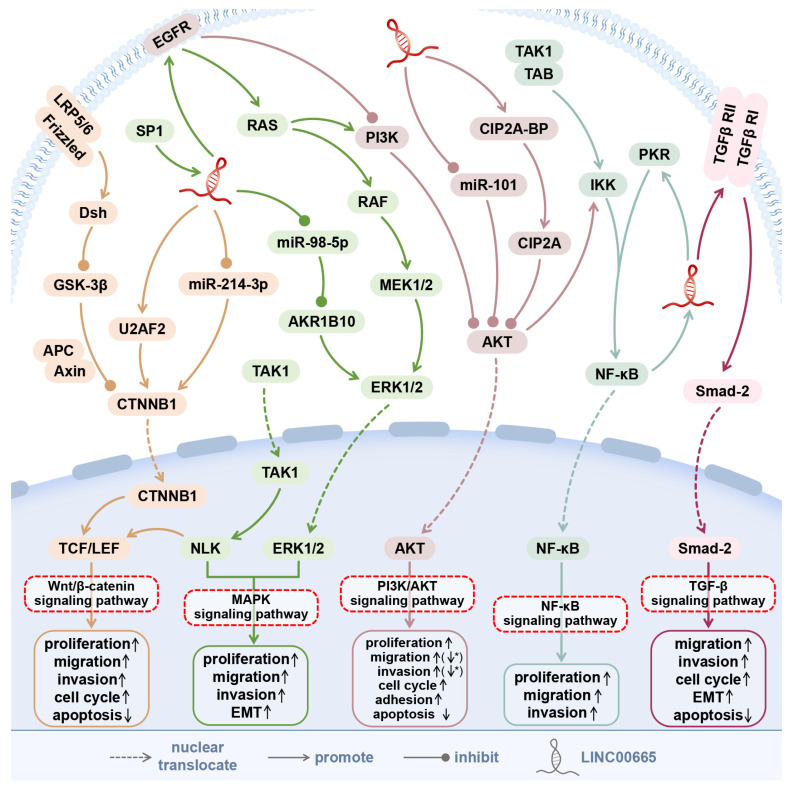
The signaling pathways involved in LINC00665. In human cancers, LINC00665 is involved in the regulation of the Wnt/β-catenin signaling pathway, TGF-β signaling pathway, NF-κB signaling pathway, PI3K/AKT signaling pathway, and MAPK signaling pathway. *: In triple-negative breast cancer, the micropeptide CIP2A-BP encoded by LINC00665 can inhibit the PI3K/AKT signaling pathway, thereby inhibiting the migration and invasion of tumor cells. ↑, Promotion; ↓, Inhibition.

**Figure 2 cells-11-01540-f002:**
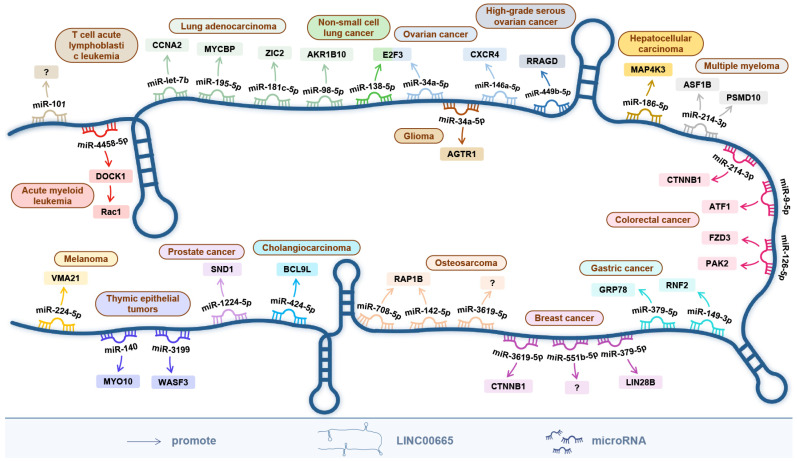
The ceRNA network centered on LINC00665. In human cancers, the ceRNA network centered on LINC00665 involves 25 miRNAs and regulates the expression of 26 mRNAs, thereby promoting tumor progression.

**Figure 3 cells-11-01540-f003:**
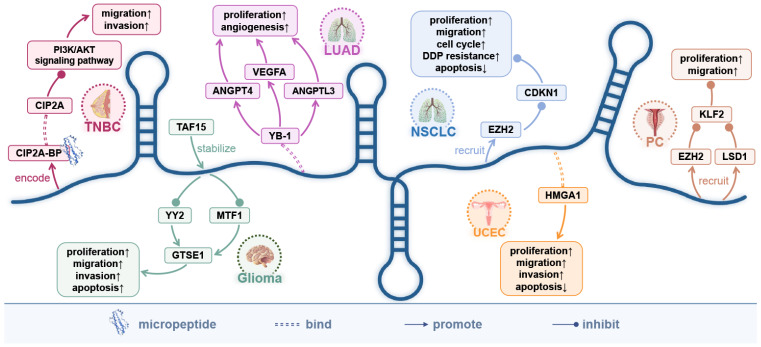
The direct targeting of LINC00665 and its encoded micro-peptide (CIP2A-BP) to protein-coding genes. The micro-peptide CIP2A-BP encoded by LINC00665 may play an important role in inhibiting the progression of triple-negative breast cancer. By directly targeting 10 protein-coding genes, LINC00665 is involved in the regulation of five cancers: TNBC, triple-negative breast cancer; LUAD, lung adenocarcinoma; NSCLC, non-small cell lung cancer; UCEC, endometrial cancer of the uterus; PC, prostate cancer. ↑, Promotion; ↓, Inhibition.

**Figure 4 cells-11-01540-f004:**
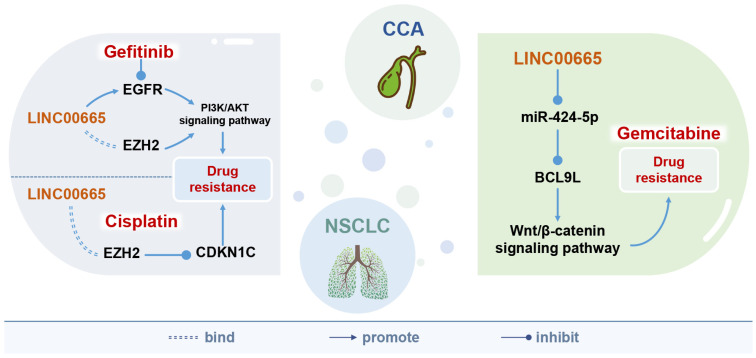
The role of LINC00665 in resistance to anticancer drugs. LINC00665 is involved in the drug action of gefitinib and cisplatin in non-small cell lung cancer, and the drug action of gemcitabine in cholangiocarcinoma, thereby enhancing tumor cell resistance. CCA, cholangiocarcinoma; NSCLC, non-small cell lung cancer.

**Figure 5 cells-11-01540-f005:**
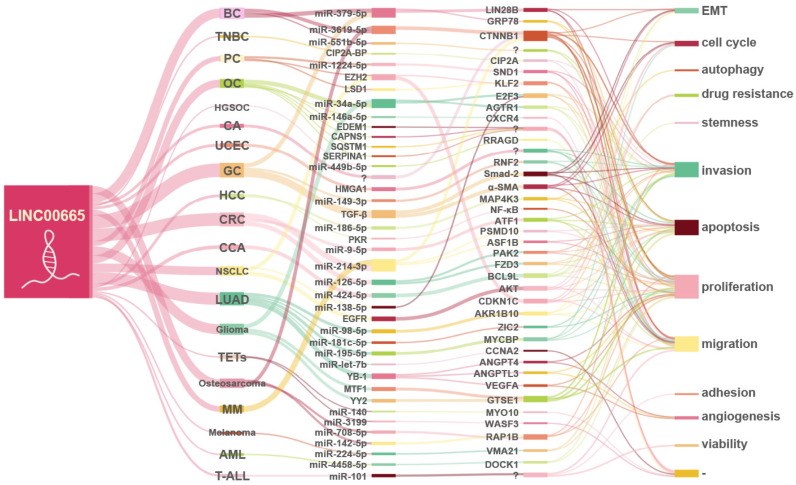
The complex molecular regulatory mechanism of LINC00665 in tumors. LINC00665 can participate in the regulation of various biological processes in 18 human tumor cells through various mechanisms, such as ceRNA, micro-peptide, and direct targeting. BC, breast cancer; TNBC, triple-negative breast cancer; PC, prostate cancer; OC, ovarian cancer; HGSOC, high-grade serous ovarian cancer; CA, cervical cancer; UCEC, endometrial cancer; GC, gastric cancer; HCC, hepatocellular carcinoma; CRC, colorectal cancer; CCA, cholangiocarcinoma; NSCLC, non-small cell lung cancer; LUAD, lung adenocarcinoma; TETs, thymic epithelial tumors; MM, multiple myeloma; AML, acute myeloid leukemia; T-ALL, T-cell acute lymphoblastic leukemia; EMT, epithelial-mesenchymal transition.

**Table 1 cells-11-01540-t001:** The role of LINC00665 in different human cancers.

Tumor Type	Assessed Cell Lines	Animals	Expression	Detection and Statistical Methods	Regulatory Mechanism	Effect In Vitro	Effect In Vivo	Ref.
Breast cancer	BC (MCF-7, MDA-MB-231, BT474, BT549, MDA-MB-468, and T47D); Normal (MCF-10A)	10 SCID mice (5-week-old, female)	Upregulation	qRT-PCR; Student’s *t*-test	LINC00665/miR-379-5p/LIN28B	proliferation↑ migration↑ invasion↑ EMT↑	tumor growth↑	[10]
	BC (MCF-7, MDA-MB-231, ZR-75-30, and MDA-MB-415); Normal (MCF-10A)	-	Upregulation	TCGA RNA-seq and qRT-PCR; Independent sample *t*-test	-	proliferation↑ migration↑ invasion↑ EMT↑	-	[11]
	BC (MCF-7 and MDA-MB-231); Normal (MCF-10A)	-	Upregulation	qRT-PCR and GEPIA; Student’s *t*-test and ANOVA	LINC00665/miR-3619-5p/CTNNB1	proliferation↑ migration↑ invasion↑ cell cycle↑ apoptosis↓	-	[12]
	-	-	Upregulation	TCGA RNA-seq and qRT-PCR; Student’s *t*-test and Chi-square test	-	cell cycle↑ DNA repair↑	-	[9]
	BC (MCF-7, MDA-MB-231, and HCC-1937); Normal (MCF-10A)	12 BALB/c nude mice (4-week-old, female)	Upregulation	TCGA RNA-seq and qRT-PCR; Student’s *t*-test, ANOVA, and Tukey post hoc tests	LINC00665/miR-551b-5p	proliferation↑ apoptosis↓	tumor growth↑	[13]
Triple negative breast cancer	TNBC (BT549, Hs578T, and MDA-MB-231); Normal (MCF-10A and HEK293T)	50 Nude mice (6–8-week-old, female)	Downregulation	qRT-PCR; two-tailed Student’s *t*-test	LINC00665/ CIP2A-BP/CIP2A	migration↓ invasion↓	tumor growth↓	[5]
Prostate cancer	PC (LNCaP, PC-3, DU-145, and 22RV1); Normal (RWPE-1)	10 BALB/c nude mice (4-week-old, female)	Upregulation	qRT-PCR; Student’s *t*-test and ANOVA	LINC00665/miR-1224-5p/SND1	proliferation↑ migration↑ invasion↑	tumor growth↑	[14]
	PC (PC-3, DU-145, 22RV1, and LNCaP)	10 Nude mice (8-week-old, male)	Upregulation	qRT-PCR; Student’s *t*-test	LINC00665/ EZH2, LSD1/KLF2	proliferation↑ migration↑	tumor growth↑	[15]
	PC (PC3)	-	Upregulation	TCGA RNA-seq and qRT-PCR; two-tailed Student’s *t*-test	-	DNA repair↑ radiosensitivity↓	-	[16]
Ovarian cancer	OC (A2780, OVCAR3, CAOV3, and SKOV3); Normal (IOSE80)	-	Upregulation	qRT-PCR; Student’s *t*-test and Pearson correlation test	LINC00665/miR-34a-5p/E2F3	proliferation↑ migration↑ invasion↑	-	[17]
	-	-	Upregulation	TCGA RNA-seq and GTEx RNA-seq	LINC00665/ EDEM1, CAPNS1, SQSTM1, SERPINA1	-	-	[18]
	-	-	Upregulation	GEO RNA-seq	LINC00665/miR-146a-5p/CXCR4	-	-	[44]
High-grade serous ovarian cancer	-	-	Upregulation	TCGA RNA-seq and GTEx RNA-seq	LINC00665/miR-449b-5p/RRAGD	-	-	[19]
Cervical cancer	CA (HeLa)	10 BALB/c immuno-deficient mice (female)	Upregulation	RNA-seq and qRT-PCR; ANOVA	LINC00665/ CTNNB1/(Wnt/β-catenin signaling pathway)	proliferation↑ migration↑ invasion↑ EMT↑	tumor growth↑	[20]
Uterine corpus endometrial cancer	UCEC (RL-95-2, Ishikawa, HEC-1B, KLE, and HHUA)	12 Mice (8-week-old, female)	Upregulation	qRT-PCR; Student’s *t*-test and ANOVA	LINC00665/ HMGA1	proliferation↑ migration↑ invasion↑ apoptosis↓	tumor growth↑	[21]
Gastric cancer	GC (MKN28, BGC-823, SGC-7901, AGS, and HGC-27); Normal (GES-1)	12 Nude mice (6-week-old, male)	Upregulation	qRT-PCR; two-tailed Student’s *t*-test and ANOVA	LINC00665/(Wnt/β-catenin signaling pathway)	proliferation↑ migration↑ invasion↑ apoptosis↓	tumor growth↑	[22]
	GC (AGS, SGC-7901, HGC27, MGC-803, MKN-45, and BGC-823); Normal (GES-1)	-	Upregulation	qRT-PCR; Student’s *t*-test and ANOVA	LINC00665/miR-149-3p/RNF2	proliferation↑ migration↑ invasion↑	-	[23]
	GC (AGS, MKN45, HGC27, MKN28, and SGC7901); Normal (GES)	BALB/c nude mice (6-week-old)	Upregulation	TCGA RNA-seq and qRT-PCR; Unpaired Student’s *t*-test and ANOVA	LINC00665/ TGF-β/Smad-2, α-SMA	EMT↑ migration↑ invasion↑ cell cycle↑ apoptosis↓	tumor growth↑	[24]
	GC (SGC-7901, AGS, and HST2); Normal (GES)	-	Upregulation	qRT-PCR; Student’s *t*-test and ANOVA	LINC00665/miR-379-5p/GRP78	proliferation↑ apoptosis↓	-	[25]
Hepatocellular carcinoma	HCC (Huh-7 and HepG2)	-	Upregulation	TCGA RNA-seq; Student’s *t*-test, ANOVA, Dunnett’s multiple comparisons test, Wilcoxon tests, and nonparametric Mann–Whitney U-test	NF-κB/ LINC00665/ PKR/NF-κB loop	proliferation↑	tumor growth↑	[26]
	-	-	Upregulation	TCGA RNA-seq, GEO RNA-seq and qRT-PCR; Independent sample *t*-tests	-	-	-	[3]
	HCC (Huh-7, HepG2, HCCLM6, MHCC-97H, and Hep3B); Normal (HL-7702)	24 BALB/c nude mice (4–6-week-old, female)	Upregulation	qRT-PCR; Student’s *t*-test and ANOVA	LINC00665/miR-186-5p/MAP4K3	proliferation↑ apoptosis↓ autophagy↓	tumor growth↑	[27]
Colorectal cancer	CRC (DLD-1, SW480, KM12, SW116, and SW620); Normal (NCM460)	-	Upregulation	qRT-PCR; Student’s *t*-test and ANOVA	LINC00665/miR-9-5p/ATF1	proliferation↑ migration↑ invasion↑ apoptosis↓	-	[28]
	CRC (SW620, LOVO, HCT-116, and SW480); Normal (NCM460)	Nude mice (Female)	Upregulation	qRT-PCR; Student’s *t*-test and ANOVA	LINC00665/miR-214-3p, U2AF2/ CTNNB1/(Wnt/β-catenin signaling pathway)	proliferation↑ migration↑ invasion↑ apoptosis↓	tumor growth↑	[29]
	CRC (DLD1, RKO, HCT116, LOVO, SW480, and NCM460)	-	Upregulation	qRT-PCR; Student’s *t*-test and ANOVA	LINC00665/miR-126-5p/PAK2, FZD3	proliferation↑ invasion↑ apoptosis↓	-	[30]
Cholangiocarcinoma	CCA (SNU-1196, SNU-1079, SNU-308, SNU-245, SNU-478, and SNU-869)	18 Nude mice	Upregulation	qRT-PCR and lncRNA microarray analysis; two-tailed Student’s *t*-test and ANOVA	LINC00665/miR-424-5p/BCL9L/(Wnt/β-catenin signaling pathway)	proliferation↑ drug resistance↑ EMT↑ stemness↑ apoptosis↓	Drug resistance↑	[8]
Non-small cell lung cancer	NSCLC (A549, H520, H1299, SPC-A1, and SK-MES-1)	24 BALB/c nude mice	Upregulation	qRT-PCR and lncRNA microarray analysis; two-tailed Student’s *t*-test and ANOVA	LINC00665/miR-138-5p/E2F3	proliferation↑ invasion↑	tumor growth↑	[31]
	LUAD (PC9)	24 Athymic BALB/c nude mice (5-week-old, male)	Upregulation	qRT-PCR; Student’s *t*-test	LINC00665/ EGFR/(PI3K/AKT signaling pathway)	proliferation↑ drug resistance↑ migration↑ cell cycle↑ apoptosis↓	tumor growth↑ Drug resistance↑	[6]
	NSCLC (PC9, SPC-A1, H1975, H1299, and A549); Normal (16HBE)	24 BALB/c nude mice (4–5-week-old, male)	Upregulation	qRT-PCR; Student’s *t*-test, Wilcoxon test, and Chi-square test	LINC00665/ EZH2/ CDKN1C	proliferation↑ migration↑ cell cycle↑ drug resistance↑ apoptosis↓	tumor growth↑ Drug resistance↑	[7]
Lung adenocarcinoma	NSCLC (A549, H1299, 16HBE, and D551)	30 BALB/c athymic nude mice (4-week-old, female)	Upregulation	qRT-PCR; Student’s *t*-test, multiple Student’s *t*-test, and nonparametric Mann–Whitney U-test	LINC00665/ YB-1/ANGPT4, ANGPTL3, VEGFA	proliferation↑ angiogenesis↑	angiogenesis↑	[32]
	NSCLC (A549, H1299, H1650, H520, SPC-A1, and SK-MES-1); Normal (16HBE)	20 BALB/c athymic nude mice (4-week-old, female)	Upregulation	qRT-PCR; Student’s t-test and nonparametric Mann–Whitney U-test	SP1/ LINC00665/miR-98-5p/AKR1B10/(ERK signaling pathway)	proliferation↑ migration↑ invasion↑ EMT↑	tumor growth↑	[1]
	LUAD (SK-LU-1 and Calu-3)	15 BALB/c nude mice (6–8-week-old, male)	Upregulation	qRT-PCR and GEPIA; two-tailed Student’s *t*-test, ANOVA and Tukey’s tests	LINC00665/miR-181c-5p/ZIC2	proliferation↑ invasion↑	tumor growth↑	[33]
	LUAD (HBE, A549, H1299, H1975, PC9, and SPC-A1)	10 BALB/c nude mice (male)	Upregulation	qRT-PCR; Student’s *t*-test and ANOVA	LINC00665/miR-195-5p/MYCBP	proliferation↑ invasion↑ cell cycle↑ apoptosis↓	tumor growth↑ metastasis↑	[34]
	-	-	Upregulation	GEO RNA-seq	LINC00665/miR-let-7b/CCNA2	-	-	[35]
Glioma	Glioma (U251 and U87); Normal (NHA)	BALB/c athymic nude mice(4-week-old)	Downregulation	qRT-PCR; Student’s *t*-test and ANOVA	TAF15/ LINC00665/ MTF1, YY2/ GTSE1	proliferation↓ migration↓ invasion↓ apoptosis↑	tumor growth↓	[43]
	Glioma (U87 MG, LN229, A172, U373, and U251); Normal (NHA)	6 Specific pathogen-free mice (4-week-old)	Upregulation	TCGA RNA-seq and qRT-PCR; Student’s *t*-test and ANOVA	LINC00665/miR-34a-5p/AGTR1	proliferation↑ migration↑ invasion↑	tumor growth↑	[36]
Thymic epithelial tumors	-	-	Upregulation	TCGA RNA-seq	LINC00665/miR-140/MYO10 & LINC00665/miR-3199/WASF3	-	-	[37]
Osteosarcoma	Osteosarcoma (143B, U2OS, MG-63, and Saos-2); Normal (hFOB1.19)	-	Upregulation	qRT-PCR; Student’s *t*-test, ANOVA, and Dunnett’s test	LINC00665/miR-3619	proliferation↑ migration↑ invasion↑	-	[38]
	Osteosarcoma (MG-63, U2OS, 143B, and Saos-2); Normal (hFOB)	-	Upregulation	qRT-PCR; Student’s *t*-test and ANOVA	LINC00665/miR-708-5p, miR-142-5p/RAP1B	proliferation↑ migration↑ invasion↑	-	[2]
Acute myelocytic leukemia	MM (MM.1S, U266, RPMI-8226, KM3, and H929); Normal (nPCs)	-	Upregulation	qRT-PCR; Student’s *t*-test and ANOVA	LINC00665/miR-214-3p/PSMD10, ASF1B	proliferation↑ apoptosis↓	-	[39]
Melanoma	Melanoma (A375, M21, A2058, and A-875)	BALB/c nude mice (6-week-old, male)	Upregulation	qRT-PCR; Student’s *t*-test and ANOVA	LINC00665/miR-224-5p/VMA21	proliferation↑ migration↑	-	[40]
Acute myelocytic leukemia	AML (KG1, U937, NB4, and HL60); Normal (HS-5)	-	Upregulation	qRT-PCR and GEPIA; Student’s *t*-test, ANOVA, and Dunnett’s test	LINC00665/miR-4458-5p/DOCK1/Rac1	proliferation↑ migration↑ adhesion↑ apoptosis↓	-	[41]
T cell acute lymphoblastic leukemia	T-ALL (MOLT-4 and CCRF-CEM); Normal (PBMC)	-	Upregulation	qRT-PCR and GEPIA; Student’s *t*-test, ANOVA, and Bonferroni post hoc test	LINC00665/miR-101	viability↑ migration↑ adhesion↑	-	[42]

SCID, Severe combined immune deficiency; EMT, Epithelial mesenchymal transformation; qRT-PCR, quantificational real-time polymerase chain reaction; TCGA, The cancer genome atlas; RNA-seq, RNA sequencing; GEPIA, Gene expression profiling interactive analysis; ANOVA, Analysis of variance; GEO, Gene expression omnibus; ↑, Promotion; ↓, Inhibition.

**Table 2 cells-11-01540-t002:** Diagnostic and prognostic value of LINC00665 in cancers.

System	Tumor Type	Sample Size	Expression	Prognostic/Diagnostic Value	Ref.
Reproductive system	Breast cancer	60 patients	Upregulation	Positively associated with lymph node metastasis and TNM stage; prognostic factor of OS (*p* = 0.0209) and DFS (*p* = 0.0492)	[11]
		106 patients	Upregulation	Positively associated with tumor size, lymph node metastasis and TNM stage	[12]
		102 patients	Upregulation	Positively associated with lymph node metastasis; AUC = 0.785	[9]
		36 patients	Upregulation	Positively associated with lymph node metastasis and TNM stage; prognostic factor of OS (*p* = 0.016)	[13]
	Triple negative breast cancer	217 patients	Downregulation	Prognostic factor of OS (MST: 29 versus 43 months, *p* = 0.02, HR = 2.64)	[5]
	Prostate cancer	50 patients	Upregulation	Positively associated with lymph node metastasis and TNM stage; prognostic factor of OS (*p* < 0.05)	[15]
		-	Upregulation	Prognostic factor of OS (*p* = 0.022)	[16]
	Ovarian cancer	56 patients	Upregulation	Positively associated with tumor size, lymph node metastasis and FIGO stage; prognostic factor of OS (*p* = 0.0037, HR = 1.37)	[17]
		-	Upregulation	Prognostic factor of OS (*p* = 5.006 × 10^−3^)	[18]
		-	Upregulation	Prognostic factor of OS (*p* = 0.00051, HR = 1.43)	[44]
	High-grade serous ovarian cancer	-	Upregulation	Negatively associated with the infiltration level of macrophages and dendritic cells	[19]
Digestive system	Gastric cancer	49 patients	Upregulation	Positively associated with poor differentiation and TNM stage	[23]
		116 patients	Upregulation	Positively associated with tumor depth, lymph node metastasis and TNM stage; prognostic factor of OS (*p <* 0.05, HR = 2.703); AUC = 0.828	[24]
	Hepatocellular carcinoma	122 patients	Upregulation	Positively associated with lymph node metastasis and TNM stage; prognostic factor of OS (*p <* 0.05)	[26]
		39 patients	Upregulation	Positively associated with poor differentiation, TNM stage and vascular invasion; prognostic factor of OS (MST: 46 versus 70 months, *p* = 0.027, HR = 1.477); AUC = 0.614, sensitivity = 0.55, specificity = 0.53	[3]
		76 patients	Upregulation	Positively associated with tumor size and poor differentiation; prognostic factor of OS (*p <* 0.05)	[27]
	Colorectal cancer	46 patients	Upregulation	Positively associated with lymph node metastasis and poor differentiation	[28]
	Cholangiocarcinoma	100 patients	Upregulation	Positively associated with lymph node metastasis and TNM stage; prognostic factor of OS (*p* = 0.0375, HR = 1.835) and RFS (*p* < 0.001, HR = 2.554)	[8]
Respiratory system	Non-small cell lung cancer	60 patients	Upregulation	Positively associated with tumor size, lymph node metastasis and TNM stage; prognostic factor of OS (*p* = 0.005) and PFS (*p* = 0.002)	[7]
	Lung adenocarcinoma	60 patients	Upregulation	Positively associated with tumor size, lymph node metastasis and TNM stage	[32]
		80 patients	Upregulation	Positively associated with tumor size, lymph node metastasis and TNM stage; prognostic factor of OS (*p* = 0.0115) and RFS (*p <* 0.001)	[1]
		84 patients	Upregulation	Prognostic factor of OS (*p* = 0.035, HR = 1.44)	[33]
		-	Upregulation	Prognostic factor of OS (*p <* 0.05)	[35]
Nervous system	Glioma	-	Downregulation	Negatively associated with pathological grade	[43]
		48 patients	Upregulation	Prognostic factor of OS (*p* = 0.0241)	[36]
Endocrine system	Thymic epithelial tumors	-	Upregulation	Prognostic factor of OS (*p* = 0.047)	[37]
Endocrine system	Osteosarcoma	33 patients	Upregulation	Prognostic factor of OS (*p <* 0.05)	[38]
		42 patients	Upregulation	Prognostic factor of OS (*p* = 0.011)	[2]

TNM, Tumor-node-metastasis; OS, Overall survival; DFS, Disease-free survival; RFS, Recurrence-free survival; PFS, Progression-free survival; AUC, Area under the curve; MST, Median survival time; HR, Hazard ratio; FIGO, Federation internationale of gynecologie and obstetrotrique.

## Data Availability

All data generated or analysed during this study are included in the article.

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
