# Peer review of "LINC00665: An Emerging Biomarker for Cancer Diagnostics and Therapeutics"

_cells, 2022, doi:10.3390/cells11091540_

Round 1
Reviewer 1 Report
This review clearly describes the impact of a long non coding RNA (LINC00665) on the regulation of signaling pathways and tumor development. The roles of the different lncRNA are not widely known in the literature and needs to be summarized through reviews like this manuscript..
The following suggestions should be taken into account to improve the manuscript:
Part 2 / table1:
Table 1 could be improved:
- do the authors provide some details about the determination of the upregulated expression of LINC00665? What is the cut-off used by the authors? Is this cut-off the same in all studies? Is it only based on fold change?
- In the same way, the “pairs of tissues” expression should by define in the text: comparison between non-tumor and tumor tissues? Using TCGA data? Cell lines or patients?
The discrepancies between the upregulation and downregulation of LINC00665 for TNBC and glioma could be discussed in the text (table 1 and 2).
Part 3:
- Typo line 93: obstetrique
- Typo line 97: low tumor differentiation, low tumor differentiation
Part 4 / table 2:
- If available in the studies, the impact on PFS and OS should be added in table 2 (i.e. number of months, the OR and p.)
Part 5:
- Figure 1 has to be improved:
- EGFR is a cell surface receptor. EGFR should be drawn in the cell surface
- The oncogenic driver RAS must be added in the figure
- The AKT pathway should be revised as followed: EGFR -> PI3K -> AKT
- The MAPK pathway should be revised as followed: EGFR -> RAS -> RAF -> MEK -> ERK and RAS -> PI3K
- Interactions between wnt and MAPK pathways could be also added
- PI3K/AKT signaling pathway promote invasion in most cancers
- 5.2: SMAD4 is often mutated in different cancers (e.g. PDAC or colorectal cancer). Is there a link between smad4 and LINC00665?
- 5.4: The PI3K/Akt pathway is not correctly described in the text (lines 194-199). PI3K don’t change the structure and activate Akt. PI3K catalyze the conversion of PIP2 into PIP3, which induce the translocation of Akt to the plasma membrane. Akt is then activated by PDK1 and MTORC2 complex.
Part 6:
- From what I know, CTNNB1 refers to the protein betacatenin coding-gene. The appropriate word should be used in part 5.2 (lines 145-166) and part 6 (line 238 or 250).
Part 8:
- Line 337: gefitinib was used in first line treatment of EGFR mutated NSCLC. This precision should be added in the text line 337.
Author Response
Response to review for LINC00665: An Emerging Biomarker for Cancer Diagnostics and Therapeutics (Manuscript ID: cells-1662789)
Dear Editor and reviewers:
Thank you for your thoughtful and useful review of our manuscript and the opportunity to further improve our work by revision. We have made all the required changes according to the suggestion by the editor and reviewers. Please find our point-by-point responses to all the inquiries below.
Reviewer 1:
- Part 2 / table1:
Table 1 could be improved:
- do the authors provide some details about the determination of the upregulated expression of LINC00665? What is the cut-off used by the authors? Is this cut-off the same in all studies? Is it only based on fold change?
Response: Thank you for your comment. We have added the Detection and Statistical Methods section to Table 1 to better show how LINC00665 expression was measured across studies. Please check the tracked changes for the details.
- On page 3, line 85, we have added the following sentences:
As shown in Table 1, qRT-PCR and RNA sequencing (RNA-seq) (PMID: 33738251, 33574708, 33767914) are commonly used to determine the expression of LINC00665 (PMID: 31907362). The analysis methods of LINC00665 aberrant expression between groups include Student's t-test (PMID: 32273723, 34094920), Analysis of variance (ANOVA) (PMID: 32983239, 33574708), and Chi-square test (PMID: 33738251). In addition, online bioinformatics analysis techniques, such as gene expression profiling interactive analysis (GEPIA) (PMID: 33658535, 34171521, 34232917), were also used to analyze the aberrant expression of LINC00665 and the correlation between LINC00665 and various parameters, such as patient baseline information, clinicopathological features, prognosis, etc.
- - In the same way, the “pairs of tissues” expression should by define in the text: comparison between non-tumor and tumor tissues? Using TCGA data? Cell lines or patients?
Response: Thank you for your comment. The "pairs of tissues" refers to matched tumor tissue and adjacent noncancerous tissue taken from a patient. However, we found that the contents of the second column (sample size) in Table 1 were not very important. In order to improve the readability of the article, we deleted the second column (sample size) in Table 1.
- The discrepancies between the upregulation and downregulation of LINC00665 for TNBC and glioma could be discussed in the text (table 1 and 2).
Response: Thank you for your comment. On page 2, line 69, we have added the following sentences:
However, in triple-negative breast cancer (PMID: 31755573), the expression of LINC00665 was significantly lower than in normal cells. In triple-negative breast cancer, activated TGF-β/Smad4 signaling significantly increases the mRNA level of 4E-BP1, which inhibits the translation of the micropeptide CIP2A-BP by binding to eIF4F (PMID: 31755573). CIP2A-BP is a micropeptide directly encoded by LINC00665 (PMID: 31755573). In the future, we need to verify whether translational repression of CIP2A-BP is the key to the reduced expression of LINC00665 (PMID: 31755573).
Furthermore, in gliomas, aberrant expression of LINC00665 has two opposing observations. Ruan et al. proposed that LINC00665 could act as a protective factor for gliomas and inhibit the malignant development of tumors through the TAF15|LINC00665/MTF1|YY2/GTSE1 axis (PMID: 32464546). Dai et al. found that LINC00665 was upregulated as an oncogenic factor and promoted glioma progression through a ceRNA mechanism (LINC00665/miR-34a-5p/AGTR1) (PMID: 33650673). These two studies used human normal astrocytes as cell line controls, which were purchased from ScienCell Research Laboratories (PMID: 32464546) and YaJi Biological (PMID: 33650673), respectively. In addition, Dai et al. found that LINC00665 (PMID: 33650673) was abnormally upregulated by analyzing 5 glioma cell lines (U87 MG, LN229, A172, U373, and U251); while Ruan et al. used only 2 glioma cell lines (U251 and U87) (PMID: 32464546). These small differences may contribute to the divergence of LINC00665 expression in gliomas. Further determination of the molecular mechanism of LINC00665 in glioma is required in the future.
- Part 3:
- Typo line 93: obstetrique
- Typo line 97: low tumor differentiation, low tumor differentiation
Response: Thank you for your comment.
(1) On page 4, line 108:
We have corrected the typo "obstetrotrigue" to "obstetrotrique".
(2) On page 4, line 112:
We have deleted the repeated "low tumor differentiation".
- Part 4 / table 2:
- If available in the studies, the impact on PFS and OS should be added in table 2 (i.e. number of months, the OR and p.)
Response: Thank you for your comment. We have added more details in Table 2 to describe the effect of aberrant expression of LINC00665 on PFS and OS.
- Part 5:
- Figure 1 has to be improved:
- EGFR is a cell surface receptor. EGFR should be drawn in the cell surface
- The oncogenic driver RAS must be added in the figure
- The AKT pathway should be revised as followed: EGFR -> PI3K -> AKT
- The MAPK pathway should be revised as followed: EGFR -> RAS -> RAF -> MEK -> ERK and RAS -> PI3K
- Interactions between wnt and MAPK pathways could be also added
- PI3K/AKT signaling pathway promote invasion in most cancers
Response: Thank you for your comment. We have modified Figure 1 to accurately show the signaling pathways involved in LINC00665. Our modifications are as follows:
- EGFR is drawn on the cell surface;
- We added the oncogenic driver RAS to the signaling pathway;
- We have modified the AKT signaling pathway. The currently upstream signaling pathway of AKT is: EGFR -> PI3K -> AKT;
- We have revised the MAPK signaling pathway. The currently MAPK signaling pathway is: EGFR -> RAS -> RAF -> MEK1/2 -> ERK1/2, and EGFR -> RAS -> PI3K;
- We have added TAK1 -> NLK -> TCF/LEF in the signaling pathway to display the interaction between the Wnt signaling pathway and MAPK signaling pathway;
- The PI3K/AKT signaling pathway promotes the invasion of most cancers, including triple-negative breast cancer. However, in triple-negative breast cancer, the CIP2A-BP micropeptide encoded by LINC00665 can inhibit the PI3K/AKT signaling pathway, thereby inhibiting the migration and invasion of tumor cells. In order to show this mechanism more clearly, we have used "↓*" in Figure 1 to indicate that the migration and invasion of cancer were inhibited after CIP2A-BP micropeptide inhibited the PI3K/AKT signaling pathway.
- The ERK signaling pathway is a part of the MAPK signaling pathway. In order to give a more comprehensive overview of the signaling pathway of LINC00665 in lung adenocarcinoma, we have replaced all the "ERK signaling pathway" in the full text with "MAPK signaling pathway".
- We have added the following information on page 7, line 224. Please check the following for the updated text:
MAPKs are serine/threonine-protein kinases. Conventional MAPKs in mammals include c-Jun NH2-terminal kinase (JNK), P38 MAPK, and extracellular signal-regulated kinase (ERK) (PMID: 35409206). As an important part of the MAPK signaling pathway, the ERK signaling pathway is a major determinant of various cellular processes that control cell proliferation, survival, differentiation, and metastasis (PMID: 28402270).
- - 5.2: SMAD4 is often mutated in different cancers (e.g. PDAC or colorectal cancer). Is there a link between smad4 and LINC00665?
Thank you for your comment. On page 6, line 182, we have added the following sentences:
However, in triple-negative breast cancer, Smad4 inhibits micropeptide synthesis of LINC00665 (PMID: 31755573). As a transcription factor for the translational repressor protein 4E-BP1, Smad4 significantly increases 4E-BP1 mRNA levels upon activation of the TGF-β/Smad signaling pathway (PMID: 31755573). 4E-BP1 inhibits translation by binding to eukaryotic translation initiation factor-4F (eIF4F), thereby reducing the expression of the micropeptide CIP2A-BP of LINC00665 (PMID: 31755573).
- - 5.4: The PI3K/Akt pathway is not correctly described in the text (lines 194-199). PI3K don’t change the structure and activate Akt. PI3K catalyze the conversion of PIP2 into PIP3, which induce the translocation of Akt to the plasma membrane. Akt is then activated by PDK1 and MTORC2 complex.
Response: Thank you for your comment. We have added the following information on page 6, line 207. Please check the following for the updated text:
The PI3K/AKT signaling pathway primarily mediates receptor-induced cell survival and is frequently altered in human cancers (PMID: 15023437). When PI3K binds to growth factor receptors such as EGFR, it can catalyze the conversion of PIP2 into PIP3, which induces the translocation of Akt to the plasma membrane (PMID: 24782981). Akt is subsequently activated by PDK1 and mTORC2 complexes, which in turn target downstream by effectors to regulate metabolism, growth, and migration in mammalian cells (PMID: 24782981). In addition, the PI3K/AKT signaling pathway also regulates the development and stability of T cells (PMID: 34171621).
- Part 6:
- From what I know, CTNNB1 refers to the protein betacatenin coding-gene. The appropriate word should be used in part 5.2 (lines 145-166) and part 6 (line 238 or 250).
Response: Thank you for your comment. We have replaced "CTNNB1" in part of the text with "CTNNB1 mRNA". Please check the following for the updated text:
(1) On page 5, line 162:
In colorectal cancer, LINC00665 can interact with U2AF2, enhance the binding between U2AF2 and CTNNB1 mRNA, and increase the stability of CTNNB1 mRNA, thereby further activating the Wnt/β-catenin signaling pathway and stimulating the proliferation and invasion of colorectal cancer cells. Among them, U2AF2 is an RNA binding protein (RBP), which has a certain binding ability to LINC00665 and CTNNB1 mRNA.
(2) On page 7, line 248:
In reproductive system tumors, LINC00665 accelerates breast cancer progression by sponging miR-379-5p, miR-3619-5p and miR-551b-5p. Among them, the LINC00665/miR-379-5p axis can upregulate LIN28B, an important regulator of EMT, while the LINC00665/miR-3619-5p axis can upregulate CTNNB1, a key protein coding-gene in the Wnt/β-catenin signaling pathway.
- Part 8:
- Line 337: gefitinib was used in first line treatment of EGFR mutated NSCLC. This precision should be added in the text line 337.
Response: Thank you for your comment. We have added the following information on page 9, line 327. Please check the following for the updated text:
Gefitinib, an epidermal growth factor receptor (EGFR) tyrosine kinase inhibitor (PMID: 30889481), has been used in the first-line treatment of EGFR-mutant non-small cell lung cancer and can significantly improve progression-free survival in patients (PMID: 35399731).
Reviewer 2 Report
-The greatest weakness of the review are in my opinion the tables 1 and 2. Its not really clear, where Lane 1 and lanes 2-end belongs to. Furthermore I suggest to use the graphical representations of the tissue as used in Figures 3 and 4 also in the Tables, to avoid confusion with the abbrevations of the tumor type.
-Table 1: Is lane 2 (sample size) really that important as an information in the table? removing it would increase the readability of the table
-Make sure the graphical representation of LINC00665 is always the same, especially in Figure 1 and 5.
-LINC00665 is upregulated in most of the cancers but downregulated in two. Maybe the authors could devote a paragraph in speculating about the reasons for that. I am aware, that in the Conclusions half a sentence is devoted to that, but one could take this conundrum to further speculate and promote about future research in the topic.
-The authors promote the idea of using LINC00665 as a future diagnostic target. However, I miss a clear statement, how they envision the use of this diagnostic marker. Do they think it is enough to use it as a single marker or do they recommend it to include it in a panel with other markers to enable differential diagnosis.
Author Response
Response to review for LINC00665: An Emerging Biomarker for Cancer Diagnostics and Therapeutics (Manuscript ID: cells-1662789)
Dear Editor and reviewers:
Thank you for your thoughtful and useful review of our manuscript and the opportunity to further improve our work by revision. We have made all the required changes according to the suggestion by the editor and reviewers. Please find our point-by-point responses to all the inquiries below.
Reviewer 2
- -The greatest weakness of the review are in my opinion the tables 1 and 2. Its not really clear, where Lane 1 and lanes 2-end belongs to. Furthermore I suggest to use the graphical representations of the tissue as used in Figures 3 and 4 also in the Tables, to avoid confusion with the abbrevations of the tumor type.
Response: Thank you for your comments. We have removed the second column of Table 1 (Sample size). At the same time, we have replaced the first column of Table 1 and Table 2 (Tumor type) with the specific name of the tumor to avoid confusion with tumor type abbreviations. Please check the tracked changes.
- -Table 1: Is lane 2 (sample size) really that important as an information in the table? removing it would increase the readability of the table
Response: Thank you for your comment. We have removed the second column of Table 1 (Sample size). Please check the tracked changes.
- -Make sure the graphical representation of LINC00665 is always the same, especially in Figure 1 and 5.
Response: Thank you for your comment. We have modified all the diagrams to ensure that the graphical representation of LINC00665 is always the same. Please check the tracked changes.
- -LINC00665 is upregulated in most of the cancers but downregulated in two. Maybe the authors could devote a paragraph in speculating about the reasons for that. I am aware, that in the Conclusions half a sentence is devoted to that, but one could take this conundrum to further speculate and promote about future research in the topic.
Response: Thank you for your comment. On page 3, line 69, we have added the following sentences:
However, in triple-negative breast cancer (PMID: 31755573), the expression of LINC00665 was significantly lower than in normal cells. In triple-negative breast cancer, activated TGF-β/Smad4 signaling significantly increases the mRNA level of 4E-BP1, which inhibits the translation of the micropeptide CIP2A-BP by binding to eIF4F (PMID: 31755573). CIP2A-BP is a micropeptide directly encoded by LINC00665 (PMID: 31755573). In the future, we need to verify whether translational repression of CIP2A-BP is the key to the reduced expression of LINC00665 (PMID: 31755573).
Furthermore, in gliomas, aberrant expression of LINC00665 has two opposing observations. Ruan et al. proposed that LINC00665 could act as a protective factor for gliomas and inhibit the malignant development of tumors through the TAF15|LINC00665/MTF1|YY2/GTSE1 axis (PMID: 32464546). Dai et al. found that LINC00665 was upregulated as an oncogenic factor and promoted glioma progression through a ceRNA mechanism (LINC00665/miR-34a-5p/AGTR1) (PMID: 33650673). These two studies used human normal astrocytes as cell line controls, which were purchased from ScienCell Research Laboratories (PMID: 32464546) and YaJi Biological (PMID: 33650673), respectively. In addition, Dai et al. found that LINC00665 (PMID: 33650673) was abnormally upregulated by analyzing 5 glioma cell lines (U87 MG, LN229, A172, U373, and U251); while Ruan et al. used only 2 glioma cell lines (U251 and U87) (PMID: 32464546). These small differences may contribute to the divergence of LINC00665 expression in gliomas. Further determination of the molecular mechanism of LINC00665 in glioma is required in the future.
- -The authors promote the idea of using LINC00665 as a future diagnostic target. However, I miss a clear statement, how they envision the use of this diagnostic marker. Do they think it is enough to use it as a single marker or do they recommend it to include it in a panel with other markers to enable differential diagnosis.
Response: Thank you for your comment. On page 4, line 142, we have added the following sentences:
The current study prefers to use LINC00665 as an independent diagnostic marker. LINC00665 is a promising diagnostic biomarker for breast cancer (PMID: 33738251), gastric cancer (PMID: 34091388), and hepatocellular carcinoma (PMID: 29728556).
Reviewer 3 Report
Comment 1.
The authors should provide a reasonable insight on why the micro-peptide CIP2A-BP encoded by LINC00665 can be expressed in only triple-negative breast cancer, but not in other cancer types. If the underlying molecular mechanism remains still not explored in previous researches, the situation should be described in the revised manuscript.
Author Response
Response to review for LINC00665: An Emerging Biomarker for Cancer Diagnostics and Therapeutics (Manuscript ID: cells-1662789)
Dear Editor and reviewers:
Thank you for your thoughtful and useful review of our manuscript and the opportunity to further improve our work by revision. We have made all the required changes according to the suggestion by the editor and reviewers. Please find our point-by-point responses to all the inquiries below.
Reviewer 3
- The authors should provide a reasonable insight on why the micro-peptide CIP2A-BP encoded by LINC00665 can be expressed in only triple-negative breast cancer, but not in other cancer types. If the underlying molecular mechanism remains still not explored in previous researches, the situation should be described in the revised manuscript.
Response: Thank you for your comment. On page 8, line 293, we have added the following sentences:
It is worth mentioning that this is the first report of CIP2A-BP. The discovery of CIP2A-BP provides a new direction for future research of LINC00665, which is expected to become a new target in cancer (PMID: 31755573).